



# Spatiotemporal variation of growth-stage specific compound climate extremes for rice in South China: Evidence from concurrent and consecutive compound events

Ran Sun[1,2,3,4], Tao Ye[1,2,3,4], Yiqing Liu[1,2,3,4], Weihang Liu[1,2,3,4], Shuo Chen[1,2,3,4]

[1]State Key Laboratory of Earth Surface Processes and Resource Ecology (ESPRE), Beijing Normal University, Beijing 100875, China
[2]Key Laboratory of Environmental Change and Natural Disasters, Ministry of Education, Beijing Normal University, Beijing 100875, China
[3]Academy of Disaster Reduction and Emergency Management, Ministry of Emergency Management and Ministry of
Education, Beijing 100875, China
[4]Faculty of Geographical Science, Beijing Normal University 100875, Beijing, China

*Correspondence to*: Tao Ye (yetao@bnu.edu.cn)

**Abstract.** There is increasing concern regarding the impact of compound agroclimatic extreme events on crop yield, particularly in the context of projected increases in their frequency and intensity due to climate change. While previous
studies have generally focused on compound hot and dry events in maize and wheat using growing-season relative thresholds, the time-variant physiological sensitivity of crops to climate extremes has not been sufficiently considered. We determined the spatiotemporal variations of compound climate extremes (CEs) for single- and late-rice in southern China during 1980−2014 and their underlying drivers using growth-stage specific physiological thresholds. Specifically, we carefully distinguished between concurrent compound events (CCEs) and consecutive compound events (CSEs). Our results
indicated an increasing trend of compound hot-dry events for single-rice, but a decreasing trend of compound chilling-rainy events for late-rice. Spatially, the hotspots of compound hot-dry events for single-rice shifted from the lower Yangtze River Basin to its upper stream, and were dominated by the spatial differences in phenology rather than the occurrence of extreme events. The hotspots of compound chilling-rainy events for late-rice remained concentrated near the northwest edges of late-rice growing areas, indicating the limitation of thermal conditions. The occurrence and duration of CCEs was closely related
to local temperature-moisture coupling (negative correlation). A path analysis suggested that temperature was the dominant factor influencing the changes in compound hot-dry events for single-rice. For the changes in compound chilling-rainy events for late-rice, the effect of temperature was only slightly larger than that of moisture. Our study has improved the understanding of compound climate extremes in China's rice production system, and the results provide important information for risk management and adaptation strategies under climate change.



## 1 Introduction

Compound climate extreme events, driven by the interaction of multiple drivers and/or hazards, often have more severe ecological and socioeconomic consequences than single events (Urban et al., 2018; Zscheischler et al., 2020). There is increasing concern regarding the future impacts of compound climate extreme events considering their projected increasing frequency and intensity (IPCC, 2022). Among the multiple potential impacts, agricultural production has received specific attention. The regional threats posed by these extreme events could further lead to global food security issues and the need to develop food system resilience (Lobell and Gourdji, 2012; Trnka et al., 2014; Chenu et al., 2017).

Previous studies have identified the spatiotemporal variation of compound agroclimatic extremes, mostly for maize and wheat. At the global and regional scales, evidence has consistently suggested an increasing frequency and intensity of compound hot-dry events during the growing season of staple crops, irrespective of the indicator and data employed in analyses (Feng et al., 2021). At the global scale, drier and hotter growing seasons have been experienced for the four major staple crops (wheat, rice, maize and soybean) from 1951 to 2020 (He et al., 2022). The researchers utilized the growing-season total precipitation and accumulated active temperature to determine the climate extremes. It has also been reported that the mean extent of global maize and wheat cropland exposed to concurrent hot-and-dry extremes has increased by ~2% over the period of 1950−2009 (Lesk and Anderson, 2021), based on growing-season standardized anomalies of soil moisture and killing-degree-days. At the regional scale, continuous upward trends have been found for compound hot-dry days for rain-fed wheat and maize in China over the period of 1980−2015. Percentiles of daily mean temperature and precipitation have been used to identify their extremes, but they tend to be calculated over a 21-day moving window instead of the whole growing-season (Lu et al., 2018). In a study focused on Chinese maize, it was found that there was a wide spatial coverage but few overall trends for compound hot-dry, hot-wet, cold-dry, and cold-wet events during 1990−2014 when using heating/freezing degree days together with standardized precipitation index over the growing season (Li et al., 2022). Similar results were also found for the impact of concurrent drought and heat events on maize production in the Huang-Huai-Hai Plain of China when using standardized precipitation evapotranspiration index and percentiles of extreme degree days from 1980−2015 (Wang et al., 2018).

Previous studies have mostly focused on staple crops, but there has been a bias toward the impact of compound hot and dry events on wheat and maize, with much less attention given to rice (Rötter et al., 2018). Most studies have used relative thresholds to define climate extremes, i.e., using percentiles of climate indicators, with less consideration given to the physiological response and thresholds of specific crops. Nevertheless, crop sensitivity to climate extremes varies by growth-stage and event type (Kern et al., 2018). Using relative thresholds instead of growth-stage specific physiological thresholds could generate substantial uncertainty in evaluation results. For example, rice is quite sensitive to high temperature stress during the heading-flowering stage (Xiong et al., 2016), but is relatively resistant to high temperature stress before the flowering stage or after pollination. In contrast, the seedling, tillering and filling stages are sensitive to low-temperature cold



damage (Guo et al., 2020). Maize is not particularly temperature sensitive during the sowing to tassel initiation stages, but is extremely sensitive to temperature in the stages from heading-flowering to grain-filling (Sánchez et al., 2014). For wheat, high temperature (>31°C) and drought stress at the start of the heading stage can result in substantial damage to floret

fertility, and reductions in grain yield. However, during the grain-filling period, the maximum temperature that the seeds can withstand is 35°C (Prasad et al., 2015; Sehgal et al., 2018).

Distinguishing growth-stage specific event types and thresholds would also enable a more detailed distinction of the temporal structures of compound events to be made (Zscheischler et al., 2020). Compound climate extremes can be further divided into concurrent compound events (CCEs) and consecutive compound events (CSEs). Here, CCEs refer to cases

where different hazards occur within the same period or growth-stage. CSEs are a series of hazards that occur sequentially in different growth stages. In agricultural production, most studies have focused on CCEs, partly due to the whole-growing-season analysis framework, and their concurrency is believed to be closely related to the strong negative coupling between temperature and precipitation (Lesk et al., 2022; Zscheischler and Seneviratne, 2017; Lesk et al., 2021; Van Den Hurk et al., 2015). In contrast, the concept of CSEs has been receiving increasing attention in other fields, such as studies of the future

frequency and impact of sequential/consecutive flood-hot events (Wang et al., 2019b; Liao et al., 2021), because such events would have more severe societal impacts than isolated extremes. The likelihood of occurrence and potential losses for crop production could be substantially different between CCEs and CSEs, and therefore further investigation is required.

China is the world's largest rice producer and has the second largest sown area (Li and Lyu, 2021). Rice production in China includes single-rice in northeast China and in the Yangtze River Basin, and double-season rice (early-rice and late-rice) in

southern parts of the country. The climate of these rice cropping systems varies substantially, from sub-tropical to warm temperate, and consequently the crop is exposed to a range of agroclimatic extremes. For single-rice, summer (July to September) is the highest temperature period in southern China and is prone to seasonal drought (Tan et al., 2020). At this time, single-rice in its jointing to flowering and maturity stage is vulnerable to the combined effects of heat and drought. From September to October each year, late-rice in its heading-flowering and grain-filling stages is critically vulnerable to

low temperatures, strong winds, and persistent rainy weather (Guo et al., 2020). These climate extremes compounded together are commonly referred to as "chilling-dew wind" and "continuous rain" events (Zhang et al., 2021; Xie et al., 2016). Therefore, focusing on the climate extremes related to rice production in China could help identify the potential differences between CCEs and CSEs.

The aim of this study was to determine the spatiotemporal variations of climate extremes for single- and late-rice in southern

China during the period of 1980−2014, and identify their underlying drivers. Unlike previous studies, we carefully distinguished between concurrent and consecutive climate extremes by specifying growth-stage physiological thresholds. We divided the rice growing season into three critical stages, the jointing-booting stage (#1), the heading-flowering stage (#2), and the grain-filling stage (#3). We considered four types of climate extremes that could substantially affect rice yield:





drought (D), heat (H), continuous-rain (R) and chilling-dew wind (C). We considered compound heat and drought events for

single-rice, and compound chilling and rainy events for late-rice. By applying growth-stage specific physiological thresholds, a rich combination of concurrent climate extremes and consecutive climate extremes were derived. We revealed the temporal trends and spatial hotspots of the occurrence and duration of climate extremes. We also analyzed the relationship between the duration of climate extremes and the variation and coupling of temperature and moisture. We were particularly interested in any underlying coupling related to consecutive events. Finally, we applied a path analysis to identify the major driving

factors of the changes in climate extremes during the study period.

## 2 Materials and Methods

### 2.1 Study area

Our study area covered the major rice growing areas in southern China, encompassing seven provinces (Fig. A1). The rice growing systems included typical double-season rice (early-rice and late-rice) in the southeast and single-season rice

(hereafter "single-rice) in the Yangtze River basin and southwestern China. Early-rice generally grows from March to July and is most frequently affected by extremely low temperatures during the seeding and nursery stages. Late-rice generally grows from July to November, and is subjected to extremely low temperatures, strong winds and persistent rainy weather in September to October. Single-rice generally grows from June to November. Its heading-flowering stages occur during the highest temperature season and are also prone to drought due to the hilly terrain of southern China (Tan et al., 2020). To best

present the complicated temporal structure of climate extremes, both single- and late-rice were considered in our analyses.

### 2.2 Data

Rice phenology data recorded by agrometeorological stations from 1981 to 2014 were obtained from the China Meteorological Administration (CMA, http://data.cma.cn). This dataset is considered the best quality crop phenology observation dataset in China (Liu et al., 2023) and has gained widespread usage (Liu et al., 2023; Zhang et al., 2022a; Chen

et al., 2021). Each station meticulously documents the rice cropping type (single-rice, early-rice, or late-rice) and the corresponding dates for every phenological event during the rice-growing season following the specifications for agrometeorological observation—Rice (QX/T 468–2018). Rigorous checks and validation during the data preparation process resulted in the production of extremely accurate data pertaining to rice phenology, with an accuracy rate exceeding 95%. We analyzed observation data from 28 stations for single-rice and 37 stations for late-rice (Fig. A1, Distribution maps

of single-rice (Shen et al., 2023) and late-rice (Pan et al., 2021) for 2020 were used). All stations had a track record of over 30 years operation. Records that exceeded twice the standard deviation were rejected to further ensure the data quality (Zhao et al., 2016).



Daily mean temperature, total precipitation and total sunshine hours were obtained from the daily dataset of basic meteorological elements of China's National Surface Weather Station (V3.0), obtained from the China Meteorological
Administration (CMA, http://data.cma.cn). Phenological information from each agrometeorological station was matched with daily weather data from the nearest surface weather station. The average distance between the agrometeorological station and the surface weather station in each match was 14 km. The 0.25° gridded daily 0−10 cm soil moisture data were obtained from the VIC-CN05.1 surface hydrology dataset (Miao and Wang, 2020). We used the daily soil moisture values for the grid cells where the agrometeorological stations were located.

**2.3 Types of compound climate extremes and their thresholds**

Three stages of rice growth that were most susceptible to extreme weather stress were considered in the study: the jointing-booting stage, the heading-flowering stage and the grain-filling stage. Here, the jointing-booting stage refers to the period from jointing to the day before heading. The heading-flowering stage refers to the period from heading to flowering, and generally lasts for 10 days. The grain-filling stage refers to the period from the 11[th] day after heading to maturity. The exact
dates of the different stages were obtained from phenological records for each year and each station.

We considered four types of climate extreme that could substantially affect rice yield: drought, heat, continuous-rain, and chilling-dew wind. Growth-stage specific thresholds for each type of climate extreme were determined according to the literature, as well as Chinese national and local standards (Table 1). These thresholds were applied to daily climate data in specific growth stages to mark the historical occurrence of certain types of climate extremes.

**Table 1 The thresholds of each type of extreme event.**

| Rice type | Growth-stage | Climate extremes | Indicator & threshold: mean daily T (℃), precipitation (*PRE*/mm),  SM (%), sunshine hours (*H*/h) | |
|---|---|---|---|---|
| **Single - rice** | Jointing-booting Heading-flowering Grain-filling | Heat | T ≥ 33 | ≥ 3 successive days <NY/T 2915-2016> |
| | | Drought | SM ≤ 75 | ≥ 15 successive days <NY/T 3043-2016> |
| **Late -rice** | Heading-flowering | Chilling-dew wind | T ≤ 20 | ≥ 3 successive days <NY/T 2285-2012> |
| | | Continuous-rain | PRE ≥ 0.1 and H ≤ 1 | ≥ 3 successive days <DB5101/T 125-2021> |
| | Grain-filling | Chilling-dew wind | T ≤ 17 | ≥ 3 successive days |



|  |  |  | < NY/T 2285-2012 > |
|  | Continuous-rain | PRE ≥ 0.1 and H ≤ 1 | ≥ 3 successive days<br>< DB5101/T 125-2021> |

**Note:** NY/T is the *Agricultural Information Resource Classification and Coding Specification* in China. DB5101/T is the *Local Standard of Chengdu, Sichuan Province*. <NY/T 2915-2016>, Identification and classification of heat injury of rice; <NY/T 3043-2016>, Code of practice for field investigations and classification of rice seasonal drought disasters in southern-China; <NY/T 2285-2012>, Technical specification of field investigations and the grading of chilling damage to rice and; <DB5101/T 125-2021>, Indica rice weather disaster level-continuous rain.

**2.4 Spatiotemporal variation of compound climate extremes**

All possible temporal compound structures of the climate extremes considered in this study are shown in Table 2. Two compound structures of climate extremes were considered: CCEs and CSEs (Zscheischler et al., 2020). The term CCE refers to the situation in which a rice crop is impacted by two types of climate extreme at the same growth-stage, i.e., simultaneous exposure to heat and drought during the jointing-booting stage of single-rice. The term CSE refers to the situation in which rice is impacted by one event at one growth-stage, and by another at a different growth-stage. Rice is affected by different stresses at various growth stages. In this study, we only considered CSEs that crossed two stages and two event types, e.g., heat at the jointing-booting stage and drought at the heading-flowering stage, which represented temperature (heat/chilling-dew-wind) and moisture (drought/continuous-rain) stresses.

**Table 2 Types of climate extremes considered in the study**

| | Heat \ Drought | #1 Jointing-booting | #2 Heading-flowering | #3 Grain-filling |
|---|---|---|---|---|
| **Single-rice** | **#1 Jointing-booting** | H1D1 (CCEs) | H1D2 (CSEs) | H1D3 (CSEs) |
| | **#2 Heading-flowering** | H2D1 (CSEs) | H2D2 (CCEs) | H2D3 (CSEs) |
| | **#3 Grain-filling** | H3D1 (CSEs) | H3D2 (CSEs) | H3D3 (CCEs) |
| **Late-rice** | **Chilling-dew wind \ Continuous-rain** | | **#2 Heading-flowering** | **#3 Grain-filling** |
| | **#2 Heading-flowering** | | C2R2 (CCEs) | C2R3 (CSEs) |
| | **#3 Grain-filling** | | C3R2 (CSEs) | C3R3 (CCEs) |





**Note:** CCEs: concurrent compound events. CSEs: consecutive compound events. H: heat. D: drought. C: chilling-dew-wind. R: continuous-rain. #1: jointing-booting stage. #2: heading-flowering stage. #3: grain-filling stage. H1D1 means that single-rice is exposed to concurrent heat and drought during jointing-booting stage (#1). H1D2 means that single-rice is exposed to heat during the jointing-booting stage (#1) and drought during the heading-flowering stage (#2).

Two indices were used to quantify the spatiotemporal variation of climate extremes: annual times of occurrence (frequency), and annual aggregate duration (duration). Here, frequency refers to the total counts of events at a site in a given year. The duration of CCEs refers to the number of days when both events occurred simultaneously, and the duration of CSEs refers to the sum of the number of days during which both hazards occurred during the two fertility periods.

**2.5 Compound events and temperature-moisture coupling**

Previous studies have shown that compound hot-dry events are closely related to the strength of single extreme weather events (Bevacqua et al., 2022; Zhang et al., 2022b; Zscheischler and Seneviratne, 2017), as well as the strength of temperature-moisture coupling (Lesk et al., 2021). We attempted to understand how the spatial differences of the occurrence and total duration of climate extremes were related to the strength of temperature-moisture coupling. Following (Lesk et al., 2021), we used the Pearson correlation coefficient for the relationship between growth-stage mean temperature ($T$) and soil

moisture ($SM$) for single-rice $r_{T,SM}$, over the study period at each station to denote the strength of coupling. Then, we plotted the station-level total duration of climate extremes over the study period against its corresponding coupling strength. We were particularly interested in how the relationship differed between CCEs and CSEs, but we also considered whether the occurrence and duration of consecutive climate extremes were related to any cross growth-stage temperate-moisture relationships. For late-rice, the major stress of chilling-dew wind was chilling (conditions were too cold), and the major

stress of continuous rain was the actual rainfall (conditions were too wet). We therefore used the correlation coefficient for the relationship between $T$ and precipitation ($PRE$) for late-rice, $r_{T,PRE}$.

**2.6 Contribution of temperature and moisture to the changes during compound events**

We attempted to understand how the temporal changes of climate extremes could be attributed to the long-term changes in temperature and moisture. Because there can be strong interactions between temperature and moisture, a path analysis was

conducted. A path analysis decomposes the interaction between the dependent and independent variables (correlation coefficients) into direct (direct path coefficients) and indirect (indirect path coefficients) on the basis of a multiple linear regression, without requiring the variables to be independent of each other (Zhang et al., 2022b). It has been widely applied to estimate the magnitude and significance of the hypothesized causal connections between the dependent and independent variables, when the effects of the variables are confounded (Zhang et al., 2022b, c; Yan et al., 2022).



We separated the system of correlations between the dependent variable and two corresponding independent variables to obtain the path coefficients. Taking single-rice as an example, the path coefficient of $T$ to the duration of climate extremes ($DUR$) $R_{T,DUR}$, which was also the Pearson correlation coefficient between $T$ and $DUR$, could be decomposed into direct and indirect effects by:

$$R_{T,DUR} = P_{T,DUR} + r_{T,SM}P_{SM,DUR} \tag{1}$$

where, $P_{T,DUR}$ is the direct path coefficient of $T$ on $DUR$, and $r_{T,SM}$ is the Pearson correlation coefficient between the two independent variables, $T$ and $SM$. Thus, $r_{T,SM}P_{SM,DUR}$ is the indirect path coefficient of soil moisture on duration. $P_{T,DUR}$ and $P_{SM,DUR}$ are two standardized linear regression coefficients obtained by regressing $DUR$ on $T$ and $SM$. An $F$-test was conducted to test the statistical significance of the results of the regression analysis, and the $p$-value for the $F$-test results was calculated. The results of the path analysis were statistically significant when the $p$-value was $< 0.01$.

Based on the direct and indirect path coefficients, and the independent variables' relative effect on the dependent variable, the determination coefficient (DC) could be derived. The DC for each climate variable is $DC_i = P_i^2$, where $i = T, SM$ or $PRE$. For the contribution from the cooperative interaction between two climate variables, the co-determination coefficient is then $DC_{co} = 2P_i r_{ij} P_j$, where $i, j = T, SM$ or $PRE$. $DC_{co}$ can indicate the extent to which the interaction of two independent variables affects the climate extremes. The total coefficient of determination ($DC_{total}$) can be obtained by summing the direct coefficients of determination and the coefficients of co-determination of all independent variables, which was used to indicate the magnitude of the joint explanatory power of temperature and moisture changes.

**3 Results**

**3.1 Temporal changes of compound climate extremes**

By applying the growth-stage specific thresholds for each type of climate extreme event, we identified the occurrence and duration of events and analyzed the temporal changes of the annual aggregate occurrence of various combinations of concurrent and consecutive climate extremes. For single-rice, the annual aggregate occurrence (Fig. 1a and 1c) and duration (Fig. A2a and A2c) both displayed an increasing trend. The frequency of CCEs increased at a rate of approximately 0.9 times/decade, which was statistically significant. CCEs during the jointing-booting stage only started to occur after 1998 (H1D1, referring to a concurrent compound hot-dry event in stage #1), while during the heading-flowering stage (H2D2) they only occurred after 2010, but at a rate of almost one every year. The heading-flowering stage, which is most vulnerable to heat damage, was exposed to more frequent compound hot-dry events. The frequency of CSEs increased at a rate of 1.1 times/decade over this period, which was also statistically significant. Some consecutive compound patterns occurred only in recent years, i.e., H1D2 and H2D1.




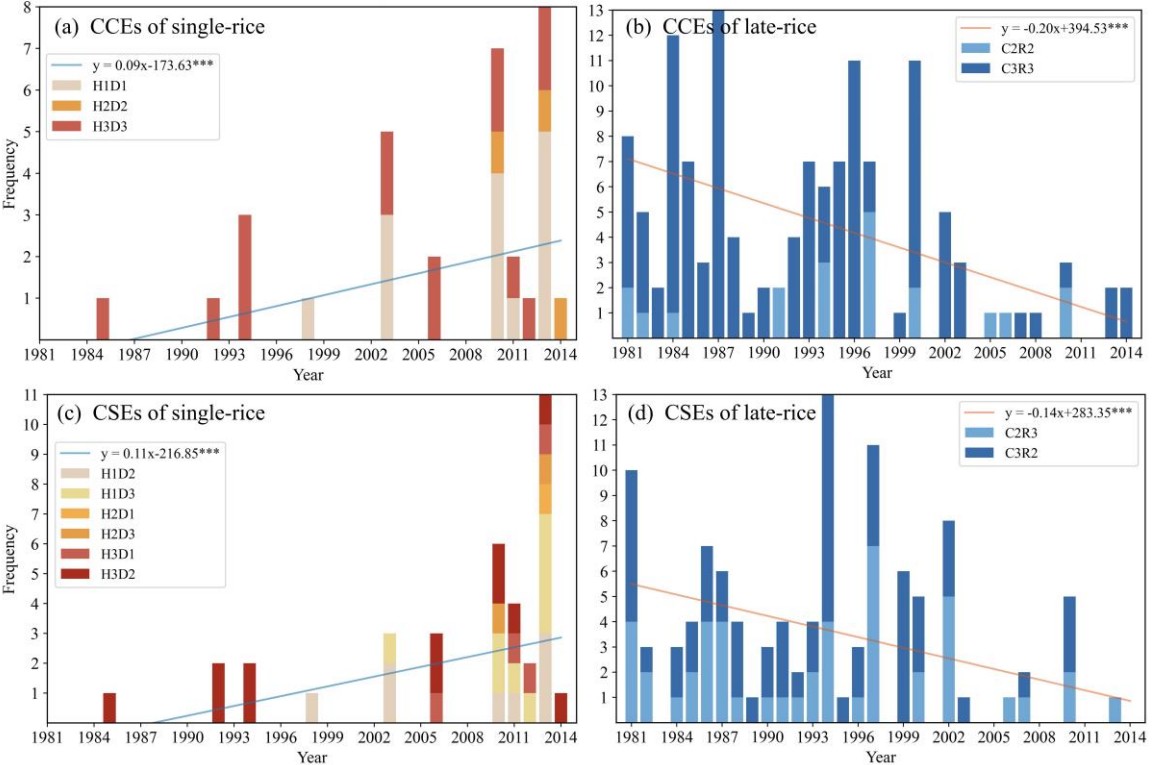

**Figure 1. Annual aggregate frequency of concurrent compound events (CCEs) and consecutive compound events (CSEs) for single- and late-rice during the period of 1981−2014.**

The changes in the annual aggregate occurrence and duration for compound events for late-rice are shown in Fig. 1b and 1d and Fig. A2b and A2d. Compound events for late-rice occurred almost every year before 2000, with two peaks of annual aggregate occurrence in 1984 and 1996. Both the frequency and duration of climate extremes decreased substantially after 2000. In particular, the frequency of CCEs decreased at a rate of 2.0 times/decade, and the frequency of CSEs decreased at a rate of 1.4 times/decade, both of which were statistically significant.

### 3.2 Spatial distribution of compound climate extremes

For single-rice, the spatial distributions of climate extremes displayed a clear pattern (Fig. 2). The number of sites that experienced compound hot-dry events during the historical period was small, but spatially concentrated. Hotspots of high-frequency sites differed by growth-stage. The hotspots of climate extremes shifted gradually from the coast to inland China along with the growth of rice. The climate extremes related to heat stress were mostly concentrated in the lower reaches of





the Yangtze River (East China region) in the jointing-booting stage (H1), and in the eastern part of the Sichuan Basin (Chongqing municipality) in the grain-filling stage (H3).

**Figure 2**. **Spatial distribution of the climate extremes of single-rice (1981−2014) based on the locations of agrometeorological stations.** Squares (a, e, i) denote concurrent compound events (CCEs) and circles (b, c, d, f, g, h) denote consecutive compound events (CSEs). The cross symbol in the middle of the pattern means that the site value was 0.

For the compound chilling-rainy events of late-rice, the high-frequency and long-duration stations were mostly located in the inland area at the northwestern edge of the growing area (Fig. 3). This pattern effectively restored the control of thermal limitations on late-rice cultivation. Overall, chilling-related events occurred in the later period of the growing season, and consequently C2R2 had the lowest frequency, only 1−2 occurrences at a few stations in Hunan (Fig. 3a). While the frequency and duration of C3R3 were very high, with most stations having more than 10 CCEs over a duration of more than





40 days in total (Fig. 3d). The spatial patterns of the two types of CSEs were relatively consistent (Fig. 3b and 3c), and were similar to that of the pattern for C3R3.

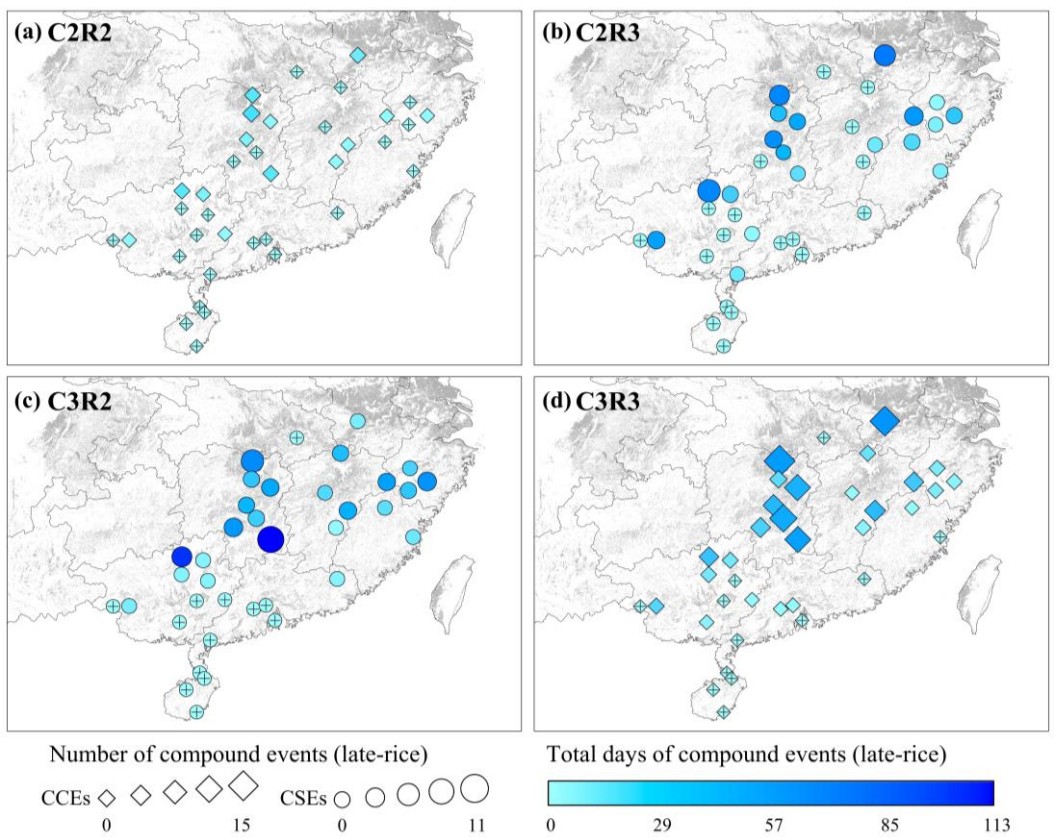


**Figure 3**. **Spatial distribution of the climate extremes of late-rice for the period of 1981−2014.** The size of the shape indicates the number of climate extremes, and the shading indicates the total duration (days). The cross symbol in the middle of the pattern means the site value is 0.

### 3.3 Dependence of compound events on temperature-moisture coupling

To determine how the spatial differences of the occurrence and total duration of climate extremes were related to the strength of temperature-moisture coupling, a composite analysis of temperature and soil moisture was conducted (Fig. 4). Our results showed that the frequency and duration of CCEs were closely related to the temperature-moisture coupling (Fig. 4a and 4c) for both single- and late-rice. Overall, the negative temperature-moisture coupling was stronger for compound hot-dry events than for compound chilling-rainy events. There was a negative relationship between duration and coupling for H2D2 and
C2R2, but it was not significant.





We were also interested in how this relationship differed for CCEs and CSEs, and whether the occurrence and duration of CSEs were related to any cross growth-stage temperature-moisture relationships for CSEs. However, no clear pattern was observed between event occurrence/duration versus growth-stage temperature-moisture coupling. The mean of the distribution of the correlation coefficient $r_{T,SM}$ was not significantly different from 0. This indicates that there was no

consistent or significant pattern of correlation between temperature and moisture across different growth stages of rice. Therefore, although CSEs were observed across growth stages over the historical period, our statistical evidence did not support any underlying climate driving force.

Figure 4. **The duration of rice climate extremes versus growth-stage temperature-moisture coupling during the**

**period of 1981−2014.** The cross sign indicates that the correlation between temperature and moisture was insignificant. The symbol * indicates that *F*-test results were significant at the 10% significance level .



### 3.4 Contribution of temperature and moisture to the changes in compound events

We took the path coefficient as the relative sensitivity of $DUR$ to $T$ and $SM$ for single-rice, and $PRE$ for late-rice. For single-rice, for all CCEs and most CSEs of the compound hot-dry type, the direct path coefficient for temperature $P_T$ was positive while the direct path coefficient for soil moisture $P_{SM}$ was negative, and the absolute values of $P_T$ were greater than those of $P_{SM}$ (Fig. 5a). This pattern indicated that temperature had a greater direct effect than soil moisture. The direct effect of soil moisture differed substantially across event types. For example, the moisture stress had a much larger effect in H3D3 than in H1D1 and H2D2, with a much higher absolute value of $P_{SM}$ that was similar to that of $P_T$. The CSEs associated with H1 and H2 events had modest $P_{SM}$ values, quite close to zero, while the CSEs associated with H3 (H3D1, H3D2) had much larger absolute values of $P_{SM}$. This suggests that for CSEs in the Sichuan Basin, soil moisture was a greater determinant of compound hot-dry events than in the Yangtze River (East China region). The influence of soil moisture on these events even approached that of temperature

For single-rice, the total determination coefficient, $DC_{total}$, which indicates the total effect of the independent variable on the dependent variable, was similar across all types of CCEs and CSEs (median around 0.4), except for H2D3 (Fig. 5c and 5d). The single-factor determination coefficients ($DC_{T,DUR}$ and $DC_{SM,DUR}$) indicated that temperature affected the changes of climate extremes to a greater extent than soil moisture, with a similar pattern observed for the path coefficients ($P_{T,DUR}$, $P_{SM,DUR}$). The median $DC_T$ was higher than the median $DC_{SM}$ for most types of CCEs and CSEs (except for H3D1). The $DC_{co}$ of temperature and soil moisture suggested a small interactive effect of temperature and soil moisture on compound hot-dry events, with $DC_{co}$ mostly < 0.1, which was much smaller than the single-factor DCs.

The pattern of the effects of temperature and precipitation on compound chilling-rainy events for late-rice was very different to that of hot-dry events for single-rice. Here, CCEs and CSEs showed a similar pattern, with negative $P_T$ values and positive $P_{PRE}$ values, and slightly larger absolute values of $P_T$ than $P_{PRE}$ (Fig. 5b). Therefore, for late-rice, both temperature and moisture had a strong direct effect on the changes in climate extremes, with temperature having a slightly larger effect than precipitation. This pattern was also supported by the DCs of individual variables. $DC_{co}$ was almost 0 for all climate extremes (Fig. 5d), which was due to the very small indirect coefficient, indicating that the interactive effects of temperature and moisture had minimal influence on the changes observed in compound chilling-rainy events for late-rice.





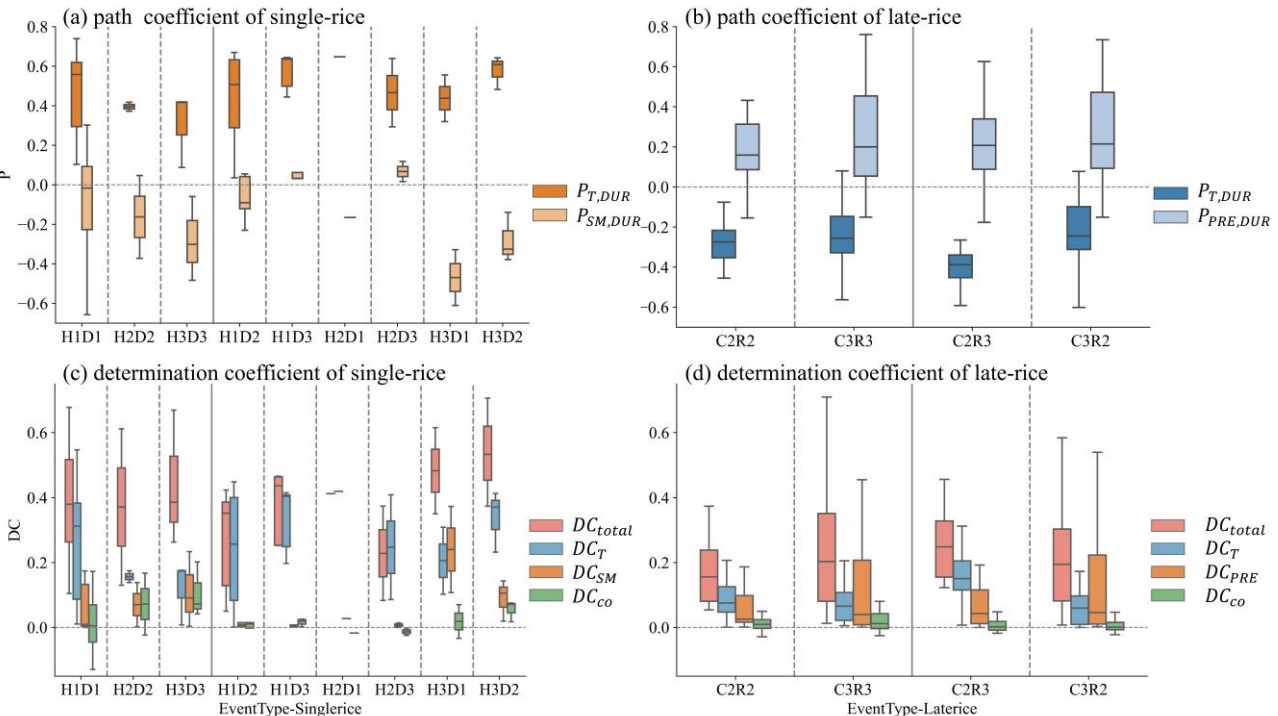

**Figure 5**. **Boxplot of the path analysis of climate factors on the duration of CCEs and CSEs for the period of 1981−2014.** *F*-test results that were statistically significant at the 0.01 significance level of were retained in the figure.

**4 Discussion**

**4.1 Spatiotemporal variation of compound events for rice in Southern China**

We attempted to reveal the spatiotemporal variation of climate extremes for single-and late-rice in southern China, using growth-stage specific thresholds for temperature and moisture (either soil moisture or precipitation). This helped to reduce the uncertainty that could have been induced using single thresholds for the entire growing season. For example, the spatial
shifts in the hotspots of compound hot-dry events of single-rice would have not been identified if we conducted evaluations using the entire growing-season. For the chilling stress to late-rice, the different effects of low temperatures at the heading-flowering and grain-filling stages would not have been distinguishable if only one single temperature threshold had been used to screen the whole growing-season. The consideration of a growth-stage specific type-threshold enabled us to distinguish the different temporal structures of CCEs and CSEs. The spatial and temporal characteristics of CCEs and CSEs
were different for single-rice.

Temporally, we found an increasing trend of compound hot-dry events for single-rice, but a decreasing trend of compound chilling-rainy events for late-rice in southern China. This result was consistent with the increasing frequency of compound





hot-dry events reported in previous studies. For example, increasing trends for compound hot-dry events in the main crop production areas since 1980 were also reported by He et al. (2022), Zhang et al. (2022c), and Lu et al. (2018), both globally

and in China. For chilling-rainy events in late-rice, Wang et al., (2019a) also reported a general decreasing trend over large parts of the single- and late-rice growing areas of China. They also suggested a large expansion of cold stress in southern parts of the mid-lower reaches of the Yangtze River during 2010−2015, despite the general warming trend. The local discrepancies between our study and their results might be due to the differences in the indicators used to represent cold stress.

Spatially, we found that concurrent hot-dry events occurred only in specific regions in each of the three growth stages of rice, and were mainly dominated by the occurrence of heat stress (H) in each growth-stage (Fig. A3). When the possible reasons for this were investigated, we found that these spatial differences were mainly attributed to the regional differences in the time of rice growth, rather than the timing of regional high-temperature events. However, high temperatures in July and August in southern China were a pre-condition for hot events, and the dates of the susceptible growth-stage eventually

determined the period of exposure. For example, the single-rice transplanting date was 30 days earlier (day of the year, DOY 174−198) in the upper-stream than in the lower Yangtze River basin (DOY 207−232). When the single-rice in Chongqing entered the grain-filling stage, rice in the middle and lower reaches of the Yangtze River had just entered the jointing-booting stage. Consequently, compound hot-dry events had a higher frequency in the later growth-stage in the upper-stream than in the lower-stream. This finding further emphasizes the importance of using growth-stage specific thresholds, which

allowed the exact spatiotemporal overlapping of climate extremes and the susceptible growth stages to be captured. In contrast, the spatial patterns of the compound chilling-rainy events for late-rice during the heading-flowering and grain-filling periods were consistent, mainly because there were no substantial differences between the growing seasons, or the distribution of chilling and rainy events (Fig. A3).

**4.2 Climate drivers of the compound events for rice in southern China**

The correlation analysis revealed that the occurrence and duration of CCEs were closely related to local temperature-moisture coupling (negative correlation). Previous studies have shown that an enhanced dry-hot dependence can lead to more frequent compound hot-dry events (Hao and Singh, 2020; Zscheischler and Seneviratne, 2017). The combination of these processes leads to a strong negative temperature-soil moisture correlation, which can be explained by two pathways: land-atmosphere feedbacks and weather-scale correspondence between clouds and incoming shortwave radiation.

Specifically, soil moisture deficits caused by low precipitation can lead to reduced evaporative cooling along with increased sensible heat fluxes and higher surface air temperatures. High temperature anomalies accelerate evapotranspiration, which further depletes soil moisture (Miralles et al., 2019; Liu et al., 2020). In addition, the low levels of cloudiness associated with low precipitation (and subsequent soil moisture deficits) tend to enhance incoming shortwave radiation, which leads to higher surface air temperatures (Berg et al., 2015). For chilling-rainy events for late-rice, our results also indicated a weak



temperature-moisture coupling. However, compared to hot-dry events, the couplings behind chilling-rainy events have largely been ignored in previous studies, and the underlying mechanism requires further investigation (Trotsiuk et al., 2020).

Our results did not provide any explicit statistical evidence that the occurrence of CSEs was related to the cross growth-stage temperature-moisture relationship. Therefore, these events, despite their historical occurrence, could be the result of a combination of independent random events. However, this did not mean that CSEs are less important because certain

patterns of CSEs could also have a strong impact on rice growth and yield. The random combination structure of these events indicates that a stochastic simulation of independent events could be an effective approach for studying CSEs. The structure of random combination suggests stochastic simulation of independent events work for CSEs. Further analysis is needed to test this hypothesis using future climate projections.

The path analysis indicated that temperature changes had a larger direct effect than moisture on the changes in most

compound hot-dry events of single-rice, but the effects of temperature and moisture changes were comparable for compound chilling-rainy events of late-rice. Previous studies of the driving factors of the changes in climate extremes have produced divergent results. For example, Zhang et al. (2022b) suggested that the changes were primarily due to the increasing temperature. In contrast, Bevacqua et al. (2022) speculated that precipitation trends are believed to determine the future occurrences of compound hot-dry events. This is because future local warming would be sufficiently large that future

droughts would always coincide with moderately hot extremes, and consequently the changes in drought frequency would become the modulating factor. In our study area, moisture variability was more likely to trigger drought events, while extremely high temperatures did not occur as frequently during the sensitive growth-stage, and therefore changes in temperature were the dominating factor for compound hot-dry events.

### 4.3 Limitations

There were several limitations to this study that could have induced uncertainties. First, our study was limited by the length of the time-series of data. Agrometeorological station data were only available up to 2014, and recent years that had experienced the most pronounced warming (IPCC, 2021) were therefore not included in the analysis. In particular, the severe compound hot-dry event in southern China in 2022 had a substantial impact on rice production (Hao et al., 2023). Second, we emphasized the importance of CSEs, but our results did not identify any clear physical driving mechanism for the

occurrence of such events. This suggests that CSEs could be treated as a random combination of single events. However, the amplification mechanism of consecutive compound events on crop yield loss still needs further exploration via controlled field experiments.





## 5 Conclusions

In this study, we investigated the spatiotemporal variation of compound climate events for single- and late-rice in southern
China and their underlying climate drivers, by distinguishing growth-stage specific event types and thresholds. We found an
increasing trend of compound hot-dry events for single-rice, but a decreasing trend of compound chilling-rainy events for
late-rice during 1980 to 2014. Spatially, the hotspots of compound hot-dry events for single rice shifted from the lower
Yangtze River Basin to its upper stream, which was dominated by the spatial differences in the growing seasons of single
rice rather than extreme events. The hotspots of compound chilling-rainy events remained concentrated near the northwest
edges of late-rice growing areas. This suggests that the limitation of suitable thermal conditions played a significant role in
determining the spatial distribution of these events. The changes in the compound hot-dry events of single-rice were
dominated by temperature for the single-rice, but the effects of temperature and moisture on the compound chilling-rainy
events of late-rice were comparable.

There are several potential avenues for further exploration and expansion of our study. The first is to include climate
projections from general circulation models (GCMs) to evaluate the future changes in the frequency and duration of CCEs
and CSEs for rice. Whether there is any physical linkage between temperature and moisture underlying the occurrence of
consecutive events should be determined in future climate projections. In addition to establishing the occurrence of crop
damage, further analysis is needed to explore the degree of rice plant damage and yield loss by compound events and to
determine their underlying mechanisms. Recent studies have provided additional details regarding the impacts of compound
events on other staple crops (Hamed et al., 2021), or single climate extremes for rice (Fu et al., 2023). It would be useful to
integrate controlled field experiments and historical observations to determine the amplification or attenuation effect when
multiple stresses are imposed, either concurrently or consecutively.





**Appendix A: Supplementary Figures**

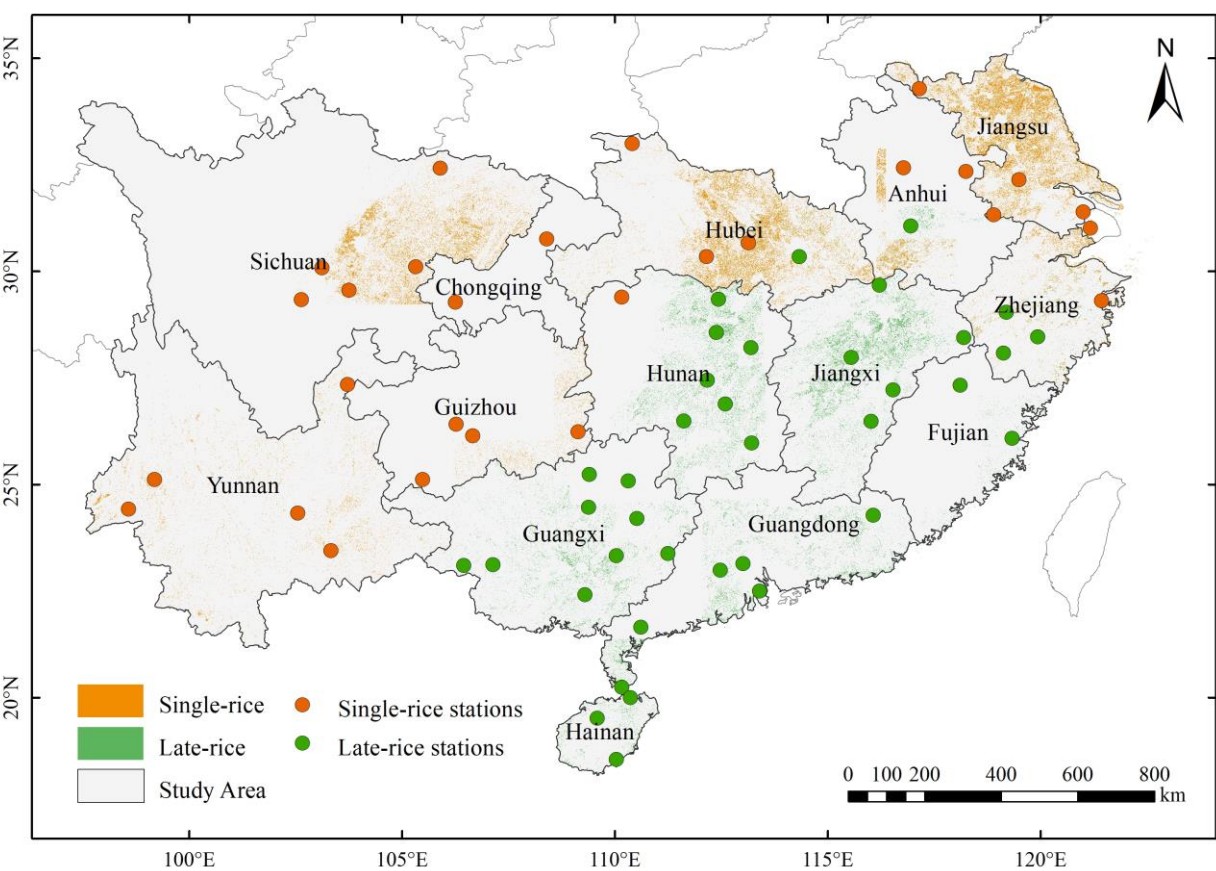

**Figure A1. The location of the agrometeorological stations.** Yellow circles indicate stations where single-rice is grown and green circles indicate stations where late-rice is grown.



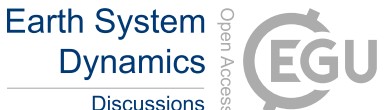

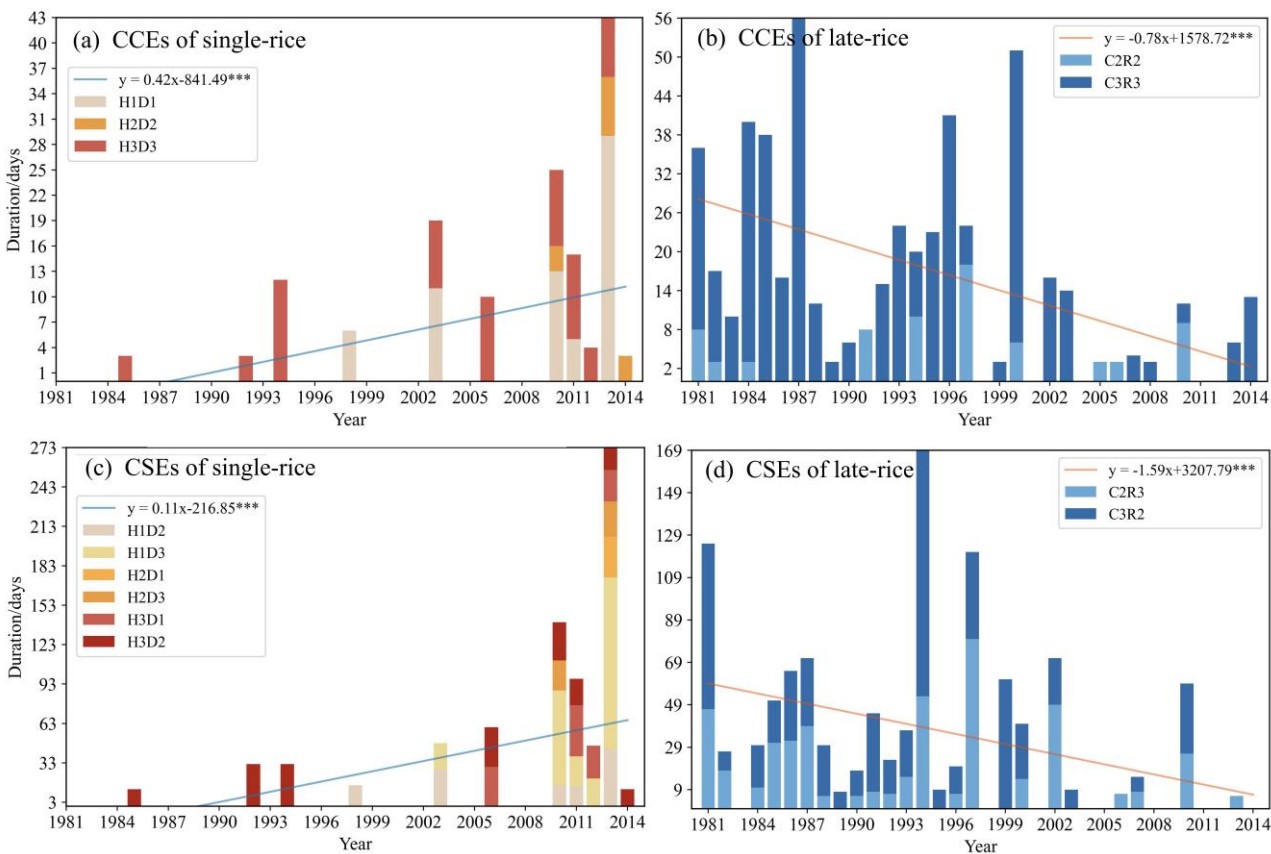

**Figure A2. Annual aggregate duration of CCEs and CSEs for single- and late-rice for the period of 1981−2014.**

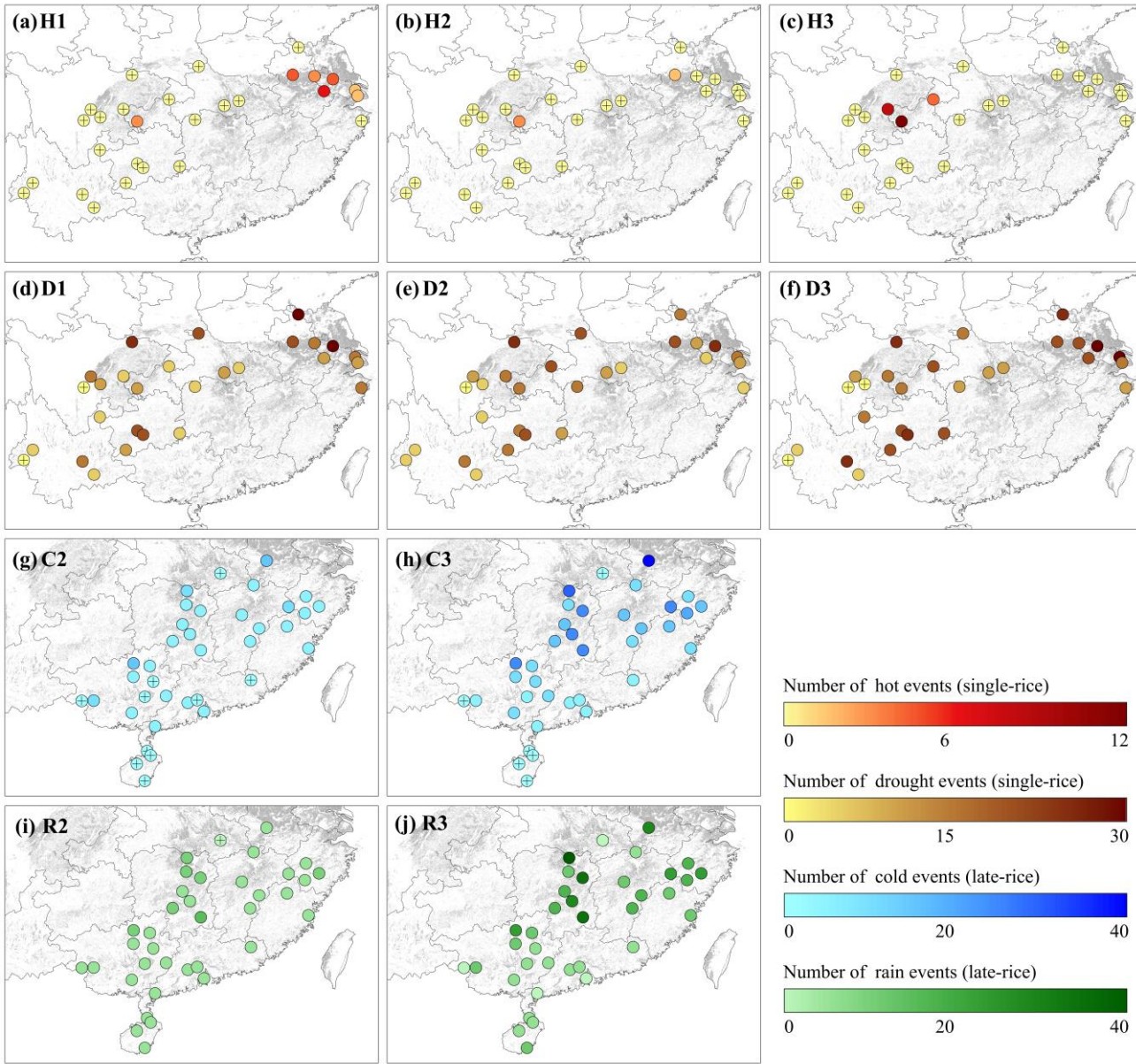

**Figure A3. Spatial distribution of single extreme events of rice for the period of 1981−2014.** Each subgraph represents the frequency of (a-c) hot events, (d-f) drought events, (g, h) cold events, and (i, j) rain events. The cross symbol in the middle of the pattern means the site value is 0.



**Author contributions**

Tao Ye designed the research. Material preparation, data collection and analysis were performed by Ran Sun, Yiqing Liu
and Weihang Liu. Ran Sun wrote the paper and Tao Ye revised it. All authors commented on previous versions of the
manuscript. All authors read and approved the final manuscript.

**Data availability**

Rice phenology data recorded by agrometeorological stations are available through the China Meteorological Administration
(CMA) at http://data.cma.cn. The daily meteorological dataset of basic meteorological elements of China National Surface
Weather Station (V3.0) are also available through the China Meteorological Administration (CMA) at http://data.cma.cn.
The 0.25° gridded daily 0-10 cm soil moisture data are available through the surface hydrology dataset VIC-CN05.1 at
https://doi.org/10.1016/j.jhydrol.2020.125413 (Miao and Wang, 2020).

**Code availability**

The code is available from the corresponding author upon reasonable request.

**Competing interests**

The authors have no relevant financial or non-financial interests to disclose.

**Acknowledgments**

This study has been financially supported by National Natural Science Foundation of China (NSFC. 42171075); State Key
Laboratory of Earth Surface Processes and Resources Ecology of China (2022-ZD-06), and the project jointly funded by
National Natural Science Foundation of China (NSFC. 72261147759) and the Bill & Melinda Gates Foundation
(2022YFAG1004).

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
