# Peer review of "Spatiotemporal variation of growth-stage specific compound climate extremes for rice in South China: Evidence from concurrent and consecutive compound events"

_Earth System Dynamics, 2024_

## Author Comment (AC1)

**Title: Spatiotemporal variation of growth-stage specific compound climate extremes for rice in South China: Evidence from concurrent and consecutive compound events**

**Response to Reviewer Comments (RC3):**

**'Comment on esd-2024-8', Benjamin Poschlod, Referee #3, 01 Jul 2024**

The study assesses the occurrence of climatic compound events during the rice growing season in China. Thereby, it relies on 34 years of data (1981 – 2014) of 65 stations across whole China. The study distinguishes between concurrent (CCE) and consecutive (CSE) events, two cropping systems (single-rice for a single harvest per season; late-rice for the second harvest of two harvests per season), and three growing stages. On this base, the authors perform a statistical exploration:

1. Plotting the frequency of event types and assessing a linear trend

2. Mapping the locations, where the events occurred

3. Performing a correlation analysis between event duration and the temperature-moisture coupling

4. Performing a "path analysis" in order to assess the contribution of temperature and moisture to the event duration.

The structure of the manuscript is clear; however, I have major concerns regarding the data, methodology, and the interpretation of the results. As the concerns are fundamental, I won't go into details with minor comments, but only raise the major concerns. Further, I have to note that I agree with the comments of the two other reviewers, where my concerns will partly overlap with.

**RE:** Thank you for your positive feedbacks on our manuscript. We are very grateful for your constructive comments and suggestions on how our manuscript can be improved. We respond to the comments and suggestions given in the text point-by-point below (in blue).

**RC3.1** Sample size

a) The whole analysis is based on 34 years and 65 stations. As the first reviewer, I think that this might be not sufficient to represent the heterogeneity of rice production areas across whole China. More importantly, the low sample size affects also the sampling of compound events. Especially for the hot & dry events (either CCE or CSE), only very few events are found. This severely limits the informative value of the following analyses.

**RE:** We have followed your suggestion by using grided data based analysis instead of station-based. We overlapped the climate data set a from CN05.1 (0.25°×0.25°) (Wu J. & Gao, 2013)and the rice distribution map, including single-rice (Shen et al., 2023) and late-rice (Pan et al., 2021) for 2020. Those datasets have been considered either as the best-quality gridded observation climate forcing dataset and the rice distribution dataset (Li et al., 2022; Yang et al., 2017; Zhu & Yang, 2020). As the climate forcing grids and rice distribution pixels differ largely

in their spatial resolution, we used climate grids with ≥5% areas of rice pixels inside. With the update, our sample sizes increased from 28 stations to 2262 grids for single-rice and from 37 stations to 1383 grids for late-rice (Fig R1). The updated sample size would be sufficient for subsequent statistical analyses.

[Figure]

Figure R1. Comparison map of station samples and updated raster samples in the study area.

b) The authors could try to interpolate the growing stage dates using climatic covariates (e.g. growing degree days) in order to better cover the whole rice production area and increase the sample size.

**RE:** We tried to interpolate phenological dates for the grids by using the annual observed dates from agrometeorological stations, as suggested by you. The stations have recorded the type of rice planted, dates of key growth stages and yield data for the period 1981-2018. This dataset is authentic and reliable for interpolation of phenology and yield. In the interpolation, we have tried different algorithms. A comparison with the high-resolution crop phenological dataset for rice in China during 2000-2019 (Luo et al., 2020) as comparison and validation dataset suggested that results from the ordinary kriging (gaussian function) be the best choice (Fig. R2). Now we have finished the interpolation of four phenological dates (booting, heading, flowering and maturity) for 34 years (1981-2018) for each of single rice and late rice. Tentatively, the interpolation results seemed to be too much smoothed, and we would continue to tune it should we have the chance to revise the manuscript.

[Figure]

Figure R2. Comparison plots of the interpolation results of the phenology dates. Taking the heading date of late-rice in 2018 as an example, (a) shows the interpolation results of this study, and (b) shows the results of resampling the ChinaCropPhen1km data to 0.25° (Luo et al., 2020).

**RC3.2** Methodology and Clarity

a) Due to the limitations of the sample size, linear trends of aggregated event frequencies (Fig. 1) and correlation analysis (Figs. 4,5) are subject to big uncertainties. Further, the trend over aggregated event types does not make any sense to me (e.g., I see an increase of H1D1 events, whereas H3D3 events do not seem to increase). The whole hot & dry analysis is based on only 1 to 6 locations (see Fig. 2).

**RE:** Following your suggestion, we will re-run the statistics based on our raster data with a significantly larger sample size.

b) The event definition nomenclature (Table 1) does not reflect the choice of thresholds intuitively: "chilling-dew wind" is based on a temperature threshold, not wind. "continuous-rain" is defined as at least three consecutive days with more than 0.1mm/d precipitation and less than an hour of sunshine. This definition includes wide ranges of precipitation (from almost dry to very wet). The sunshine threshold is more specific and might dominate this event definition. So, it's more "cloudiness" than "continuous rain".

**RE:** Thank you for raising this question. The two terms are in local Chinese context. Chilling-dew-wind is a kind of meteorological phenomenon that occurs in the area south of the Yangtze River around the Cold Dew Festival (Oct 8 or 9). Chilling-dew-wind is a cold damage that reduces rice production due to significant cooling caused by cold air invasion in autumn, which is very harmful to crop production. Chilling-dew-wind occurs during the critical period from heading to grain filling of late rice in southern China. For this reason, it is the main agrometeorological disaster in the late growing stage of late-rice production. Considering that the term Chilling-dew-wind has obvious regional characteristics and is not easy to be understood, we plan to replace it with the more understandable "chilling" for simplicity.

For the "continuous-cloudiness-rainy" event, in the local Chinese context, refers to both "cloudiness" than "continuous rain", each of which have specific way of influencing rice yield. In In southern China, continuous rain could cause pollen grains to break and anthers to be washed away by the rain, affecting fertilization and filling, which in turn can lead to yield loss (Tian & Huo, 2019). Continuous cloudiness would affected the photosynthesis of rice, leading to a decrease in the number of tillers, a decrease in the accumulation of dry matter, and a decrease in the fruiting rate (Zhang & Zheng, 2017). The standard we followed in the previous version seemed to use wide ranges of precipitation but very narrow range of sunshine. In the revision, we used precipitation for two reasons. Firstly, there is a clear correlation between daily precipitation and sunshine hours, and therefore using precipitation could also represent the occurrence of cloudiness, partly (Fig. R3). Second, we tested both indicator in the yield impact analyses, and continuous-rain severity derived much more apparent yield impact than cloudiness. Please refer to the response to comment RC3.3 for more details (Fig. R4 & R5).

[Figure]

Figure R3. Scatter plot of daily. sunshine duration and precipitation for stages #2 and #3 for late rice.

c) I could not well follow the methodological description in L177-201 and the respective results (Fig.5).

Fig. 5: For the event type H2D1, there is only one event at one location. How can there be a meaningful correlation or "path analysis" between event duration and climate drivers?

**RE:** Thank you for pointing this out. Our previous results have indeed suffered from limited sample size. The entire study would benefit from the change to a raster-data based analysis covering all rice cultivation areas in southern China. However, due to the limited time to prepare this response letter, we have not managed to re-run all our analyses on the revised gridded input data. We will try to show this in the formal revision stage.

**RC3.3** Relation to impact

a) I acknowledge the application of plant-specific absolute thresholds, which are guided by literature (Tab. 1), as well as the separation into three growing stages and two cropping systems. However, the added value is not proven, as there is no assessment of the impact variable (yield). The motivation for the authors' thresholds comes from literature, which considers the climate driver univariately (e.g., T >= 33°C is harmful for rice, independently from the moisture conditions). However, when jointly occurring with dry soil conditions, this temperature threshold could be at lower temperature.

**RE:** Thank you for your suggestions. We have tried to assessed the actual impact of climate indicators on yield (Fig. R4). Tentatively, we have firstly finished the evaluation of late rice against compound chilling-rainy events.

Here, we used AsiaRiceYield4km data (H. Wu et al., 2023) as the yield raster data, covering the period of 1995 to 2015. It is so far the dataset that provides the longest time-series covering whole China rice cultivation areas. Rice yield data with even longer time-series could only rely on the agrometeorological stations, which would again suffer from the sample size issue. To measure the impact, we followed Ye (Ye et al., 2015) by using detrended yield anomaly to remove the spatial difference in yield.

For the intensity of events, we used severity indicators based on suggestion- RC3.3 (b). For chilling, we used the cold-degree-days of the growth stage based on the concept of severity. The cumulative deficit of average daily temperature (Tmean) $\leqslant 20°C$ for three or more consecutive days:

$$CDD_{stage} = \sum_{i=1}^{n} |TEM_{base} - TEM_i|$$

$CDD_{stage}$ represents the cold-degree-days for each growth stage. $i$ is the index of the day within the consecutive days that meet the condition. $TEM_i$ is the mean daily temperature value on day $i$. $TEM_{base}$ is the mean daily temperature threshold (20°C during Heading-flowering stage (stage#2) and 17°C during Grain filling stage (stage#3), according to our threshold indicated in the manuscript. $n$ is the number of consecutive days that satisfy the condition (at least 3 days).

For the impact of rainy event, we used the cumulative precipitation greater than or equal to 25 mm for three or more consecutive days. A daily 25mm rainfall was classified as the rainy in <QX/T, 468-2018, Code of Agricultural Meteorological Observations-Rice> for precipitation:

$$PDD_{stage} = \sum_{i=1}^{n} |PRE_i - PRE_{base}|$$

$PDD_{stage}$ represents the precipitation-degree-days for each growth stage. $i$ is the index of the day within the consecutive days that meet the condition. $PRE_i$ is the daily precipitation value on day $i$. $PRE_{base}$ is the daily precipitation threshold (25 mm). $n$ is the number of consecutive days that satisfy the condition (at least 3 days).

[Figure]

Figure R4. Late-rice yield responses to severity of chilling (temperature) and rainy (precipitation) variation. Color bands indicate the value of the yield anomaly.

Several interesting things could be observed from the figures:

1) There is a clear compound impact of chilling-rainy events on late rice. As severity of chilling or rainy events increased (from the bottom left to the top right of the graphs), yield decreased. The scatters indicate a weakly concave set of isolines, indicating a larger impact on yield than the linear average of single events, that said, the compound impact of having chilling-rainy together would be stronger than the linear combination of the impacts from each stressor.

2) The impact was more severe the Heading-flowering stage (stage#2) than in the Grain filling stage (stage#3), although there were much less compound events in stage #2 than in stage #3 stage. Negative yield anomaly occurred at smaller values of severity in Fig. R4(a) than that in Fig. R4(b).

[Figure]

Figure R5. Late-rice yield responses to severity of chilling (temperature) and cloudy (sunshine hours) variation. Color bands indicate the value of the yield anomaly.

We have also tried to use the concept of continuous cloudiness, by using a severity indicator of cumulative sunshine deficit $\leqslant$ 5h. As shown in Figure R5, the is also certain pattern of the concurrent impact should we use sunshine hours to denote deficit in solar radiation. However, the pattern was much less clear than the case in Figure R4, particularly for the stage #2. Therefore, tentatively we have decided to use chilling and rainy events for late rice.

Due to the limited time of writing up this response, we have not yet finished the rest part of the analyses, i.e. the impact of concurrent heat-drought events on single rice, and the consecutive events.

b) As the first reviewer comments, the event intensity is not considered in this study. It might be useful to apply bivariate event definitions, which consider the intensity of the marginals. This could be implemented, e.g. via copulas. See Zscheischler et al., 2017 for an application and Salvadori et al., 2016 for the theory. As a starting point, the authors could use their univariate thresholds for the marginals, and apply survival Kendall return periods to assess the bivariate occurrence probability. That probability would then ideally show a higher correlation with the yields than the correlation between each marginal and the yield.

RE: Thank you for your suggestion. In the revision, we plan to use an indicator that combines both the intensity and duration of the occurrence of an extreme event: severity (Haqiqi et al., 2021) for each of the climate factor (Fig. R6). According to this reference, we define severity here base on the cumulative deviation from the threshold value of each hazard. We have exactly done so in the example of evaluating yield impact. Then, we could follow your approach to derive the bivariate probability as the measure of intensity of the compound event, by using copulas and survival Kendall return periods approach or similar approaches. An example of computing severity for chilling and rainy events have been supplied in the response to RC 3.3 (a). We would also apply this to heat and drought events.

[Figure]

**Figure 1.** Soil moisture dynamics within a typical growing season. Some soil moisture conditions can be harmful to crops, including excess wetness (i), moisture stress intensity (ii), duration of moisture stress (iii), and severity of soil moisture stress (iv). The around normal (*) levels can be determined by statistically examining the impacts of various intervals of soil moisture deviation from normal (seasonal mean volumetric soil moisture).

Figure R6. Reference chart for the definition of severity (https://doi.org/10.5194/hess-25-551-2021).

**RC3.4** Analysis & interpretation of the results

a) I cannot follow some of the interpretations. In section 3.3 (L244ff) the authors claim to show the "dependence of compound events on temperature-moisture coupling". The event itself is defined via the joint exceedance of temperature and moisture thresholds. As far as I understand, the "temperature-moisture coupling" is the Pearson rank correlation between temperature and moisture during the growing phase (see L165-176). By definition of a bivariate event, the event occurrence will be dependent on the marginal probabilities and the joint dependence structure. So, I do not see the informative value of section 3.3. and Fig. 4.

**RE:** Following your suggestion, we will redefine compound events through the marginal probabilities and the joint dependence. And explore relationships between event duration versus the temperature-moisture correlation on that basis.

We intended to link the likelihood/severity of compound events to climatological coupling between temperature and moisture. Our hypothesis follows your comment: locations/stations with stronger temperature-moisture coupling could have much more frequent/severe compound events. Our existing results based on limited sample showed that, there is some weak evidence supporting that hypothesis, for concurrent compound events. But for consecutive events, there seemed no linkage between the correlation and the CSEs. With the update in the input data, we will re-run this part of analyses to check whether there would be strong evidence rejecting above hypothesis. The test will help us decide whether to keep this part of analyses, or focus on yield impact exclusively.

Further, regarding Fig. 4: I do not consider it appropriate to assess linear relationships between event duration (total number of event days) on the y-axis versus the temperature-moisture correlation on the x-axis. The kernel density estimates suggest nicely distributed data – in reality

there is so few data, that a histogram is more appropriate. Furthermore, this whole analysis again suffers from the sampling. Taking the example of the H1D1 event, 6 locations show events at all. 5 of them are clustered in the north east (see Fig. 2a). By that means, the analysis is sensitive to the spatially inhomogeneous sampling density of locations. b) Section 3.4 claims to assess the "contribution of temperature and moisture to the changes in compound events". I do not see how the performed analysis incorporates *changes* in compound events. For the hot & dry part, this analysis shows a large amount of variability (Figs. 5a,c), which I'd attribute to the low number of sampled events. I would be very careful to (over-)interpret these results.

**RE:** Thank you for your suggestion. In the current manuscript, we intended to show that, stations with stronger temperature-moisture correlation in climatological mean are more likely to experience compound events. Here the correlation was based on climatological conditions (differ by location/station with multi-annual average condition), while the total duration of events was used to denote the overall likelihood/duration/intensity of the location/station). But yes, our current results still suffered from the limited sample size. We will update the figures after re-running all analyses by using the updated input data.

**References:**

Haqiqi, I., Grogan, D. S., Hertel, T. W., & Schlenker, W. (2021). Quantifying the impacts of compound extremes on agriculture. *Hydrology and Earth System Sciences*, *25*(2), Article 2. https://doi.org/10.5194/hess-25-551-2021

Li, Z., Liu, W., Ye, T., Chen, S., & Shan, H. (2022). Observed and CMIP6 simulated occurrence and intensity of compound agroclimatic extremes over maize harvested areas in China. *Weather and Climate Extremes*, *38*, 100503. https://doi.org/10.1016/j.wace.2022.100503

Luo, Y., Zhang, Z., Chen, Y., Li, Z., & Tao, F. (2020). ChinaCropPhen1km: A high-resolution crop phenological dataset for three staple crops in China during 2000–2015 based on leaf area index (LAI) products. *Earth System Science Data*, *12*(1), 197–214. https://doi.org/10.5194/essd-12-197-2020

Pan, B., Zheng, Y., Shen, R., Ye, T., Zhao, W., Dong, J., Ma, H., & Yuan, W. (2021). High Resolution Distribution Dataset of Double-Season Paddy Rice in China. *Remote Sensing*, *13*(22), Article 22. https://doi.org/10.3390/rs13224609

Shen, R., Pan, B., Peng, Q., Dong, J., Chen, X., Zhang, X., Ye, T., Huang, J., & Yuan, W. (2023). High-resolution distribution maps of single-season rice in China from 2017 to 2022. *Earth System Science Data*, *15*(7), 3203–3222. https://doi.org/10.5194/essd-15-3203-2023

Tian, J., & Huo, Z. (2019). Spatial-temporal Variation and Zoning of Rain-washing Damage to Early Rice Pollen in Jiangxi Province. *J Appl Meteor Sci*, *30*(5), 608–618. https://doi.org/10.11898/1001-7313.20190509

Wu, H., Zhang, J., Zhang, Z., Han, J., Cao, J., Zhang, L., Luo, Y., Mei, Q., Xu, J., & Tao, F. (2023). AsiaRiceYield4km: Seasonal rice yield in Asia from 1995 to 2015. *Earth System Science Data*, *15*(2), 791–808. https://doi.org/10.5194/essd-15-791-2023

Wu J., & Gao X. (2013). A gridded daily observation dataset over China region and comparison with the other datasets. *Chinese Journal of Geophysics*, *56*(4), 1102–1111. https://doi.org/10.6038/cjg20130406

Yang, F., Lu, H., Yang, K., He, J., Wang, W., Wright, J. S., Li, C., Han, M., & Li, Y. (2017). Evaluation of multiple forcing data sets for precipitation and shortwave radiation over major land areas of China. *Hydrology and Earth System Sciences*, *21*(11), 5805–5821. https://doi.org/10.5194/hess-21-5805-2017

Ye, T., Nie, J., Wang, J., Shi, P., & Wang, Z. (2015). Performance of detrending models of crop yield risk assessment: Evaluation on real and hypothetical yield data. *Stochastic Environmental Research and Risk Assessment*, *29*(1), 109–117. https://doi.org/10.1007/s00477-014-0871-x

Zhang H., & Zheng H. (2017). GIS-based risk assessment of the impact of continuous rain in autumn on agricultural production: A case study of the basin area in Sichuan Province, China. *Chinese Journal of Applied Ecology*, *28*(8), 2569. https://doi.org/10.13287/j.1001-9332.201708.027

Zhu, Y.-Y., & Yang, S. (2020). Evaluation of CMIP6 for historical temperature and precipitation over the Tibetan Plateau and its comparison with CMIP5. *Advances in Climate Change Research*, *11*(3), 239–251. https://doi.org/10.1016/j.accre.2020.08.001

---

## Author Comment (AC2)

**Title: Spatiotemporal variation of growth-stage specific compound climate extremes for rice in South China: Evidence from concurrent and consecutive compound events**

**Response to Reviewer Comments (RC2):**
**'Comment on esd-2024-8', Anonymous Referee #2, 30 Jun 2024**

The text is well structured. The flow is generally clear. Considering phonologically relevant growth stages to assess climatic conditions on crops is indeed interesting and adds to the value of paper. However, I have three major comments:

**RE:** Thank you for your comments and suggestions. We have responded to the comments and suggestions given in the text point-by-point below (in blue).

**RC2.1** You have not directly evalauted how/if your climatic indicators actually impact the yields: I would have expected to see some crop simulation with climatic indicators or at least a correlation analysis between crop and climatic conditions. Check following papers to get some inspiration (Luan et al., 2021; Zhu & Troy, 2018; Zscheischler et al., 2017). The way you presented the result in current version, we cannot even be sure even your indicators matter for crops and impact them.

**RE:** Thank you for your suggestions. We have preliminarily conducted yield impact analyses by using correlation between historical yield and the severity of climatic indicators. Due to the limited time, however, we have not finished all the analyses, but preliminary results on compound chilling-rainy events on late rice are presented here.

Here, we used AsiaRiceYield4km data (Wu et al., 2023) as the yield raster data, covering the period of 1995 to 2015. It is so far the dataset that provides the longest time-series covering whole China rice cultivation areas. Rice yield data with even longer time-series could only rely on the agrometeorological stations, which would again suffer from the sample size issue. To measure the impact, we followed Ye (Ye et al., 2015) by using detrended yield anomaly to remove the spatial difference in yield.

For the intensity of events, we used severity indicators based on suggestion-RC1.4. For chilling, we used the cold-degree-days of the growth stage. The cumulative deficit of average daily temperature (Tmean) $\leqslant$20°C for three or more consecutive days:

$$CDD_{stage} = \sum_{i=1}^{n} |TEM_{base} - TEM_i|$$

$CDD_{stage}$ represents the cold-degree-days for each growth stage. $i$ is the index of the day within the consecutive days that meet the condition. $TEM_i$ is the mean daily temperature value on day $i$. $TEM_{base}$ is the mean daily temperature threshold (20°C during Heading-flowering stage (stage#2) and 17°C during Grain filling stage (stage#3), according to our threshold

indicated in the manuscript. $n$ is the number of consecutive days that satisfy the condition (at least 3 days).

For the impact of rainy event, we used the cumulative precipitation greater than or equal to 25 mm for three or more consecutive days. A daily 25mm rainfall was classified as the rainy in <QX/T, 468-2018, Code of Agricultural Meteorological Observations-Rice > for precipitation:

$$PDD_{stage} = \sum_{i=1}^{n} |PRE_i - PRE_{base}|$$

$PDD_{stage}$ represents the precipitation-degree-days for each growth stage. $i$ is the index of the day within the consecutive days that meet the condition. $PRE_i$ is the daily precipitation value on day $i$. $PRE_{base}$ is the daily precipitation threshold (25 mm). $n$ is the number of consecutive days that satisfy the condition (at least 3 days).

[Figure]

Figure R1. Late-rice yield responses to severity of chilling (temperature) and rainy (precipitation) variation. Color bands indicate the value of the yield anomaly.

Several interesting things could be observed from the figures:

1) There is a clear compound impact of chilling-rainy events on late rice. As severity of chilling or rainy events increased (from the bottom left to the top right of the graphs), yield decreased. The scatters indicate a weakly convex set of isolines, indicating a weakly stronger yield impact than the linear average of single events, that said, the compound impact of having chilling-rainy together would be stronger than the linear combination of the impacts from each stressor.

2) The impact was more severe the Heading-flowering stage (stage#2) than in the Grain filling stage (stage#3), although there were much less compound events in stage #2 than in stage #3 stage. Negative yield anomaly occurred at smaller values of severity in Fig. R1(a) than that in Fig. R1(b).

Due to the limited time of writing up this response, we have not yet finished the rest part of the analyses, i.e. the impact of concurrent heat-drought events on single rice, and the consecutive events.

**RC2.2** The text is rather clear when you generally talk about compound heat and drought and the temperature moisturize coupling, in relation to these two indicators. The text, however, becomes vague when you talk about chilling and rain events and how you tried to associate them to some underlying climatic contributor. L174-176 is very unclear and requires further explanation of the method.

**RE:** Thanks for the suggestion. The major stress of compound chilling-rainy events was chilling (conditions were too cold), and the actual rainfall (conditions were too wet), we therefore used the Pearson correlation coefficient for the relationship between growth-stage mean temperature (T) and s precipitation (PRE) for late-rice $r_{T,PRE}$, over the study period at each station to denote the strength of coupling. Then, we plotted the station-level total duration of compound chilling-rainy events over the study period against its corresponding coupling strength. We have expanded this section in the manuscript.

**RC2.3** It is unclear to me why you considered two event types for CSE, according to L152-154. Why two drought or two heat within two growth stage is not considered a consecutive event? In the same lines L153-154 is unclear and requires clarification.

**RE:** Thank you for your suggestion. Two drought or two heat within two growth stage could also be regarded as a type of consecutive event if we relax the assumption of CSE. In the current version of the manuscript, we excluded such types for two reasons. Firstly, we wanted to link the severity of compound events to the coupling of different climate factors. So at the beginning we requested that the event should consists of two distinct climate factors, i.e. temperature and moisture. Two drought or two heat within two growth stage would be a matter of time-series of moisture, or temperature. Secondly, if we add two droughts or two heats within two growth stage as a consecutive compound event, the types of compound events could be too much complicated. So tentatively we stick to our structure, but will try to analyze the severity, and yield impact of such events.

**RC2.4** Specific Comment

Abstract: Consider removing the part talking about maize and wheat. The paper focuses on rice and that needs to be brought up in the abstract.
L62-66: Again consider removing the part talking about wheat and maize, and their growths temperature dependent thresholds. I think they distract the reader.

**RE:** Thank you for your suggestion. We have removed the parts mentioned by you to keep the manuscript focused.

L93: grain-filling and everywhere when you mention this word: consider removing the dash line between grain and filling. For your other stages the dash bridges two stage but grain filling is a

distinct stage itself.

**RE:** Thank you! We have revised as suggested, replacing "grain-filling" with "grain filling".

L95: I don't understand why use the term 'dew' sometimes after chilling. Maybe be consistent and use the same terminology or be specific why you need to mention dew in specific parts of the text.

**RE:** Chilling-dew-wind is a kind of meteorological phenomenon that occurs in the area south of the Yangtze River around the Cold Dew Festival (Oct.8 or 9). Chilling-dew-wind is a cold damage that reduces rice production due to significant cooling caused by cold air invasion in autumn, which is very harmful to crop production. Chilling-dew-wind occurs during the critical period from heading to grain filling of late rice in southern China. For this reason, it is the main agrometeorological disaster in the late growing stage of late-rice production. The term Chilling-dew-wind has a quite Chinese context, but its main way of affecting rice yield is "chilling". So in the rest part of the manuscript, we have used "chilling".

L115: be consistent and use either early rice or single rice. Also, here in L115 it feels like you have three type of rices while I assume there are two rices analyzed in this study.

**RE:** Thank you. For clarification, we did analyze two types of rice, the single-season rice (single-rice), and the late-season rice (which is second period of the double-season rice, early-rice and late-rice). We will delete "early-rice" at L115 to avoid unnecessary misunderstanding.

Fig 2 &3 : It is unclear to me how you considered total days of compound event. Is it the total during study period? – According to L160-163 they should correspond to yearly values but then did you consider an average of duration per year, over the study period and plotted them in these figures?

**RE:** Here, the duration of concurrent compound events (CCEs) refers to the number of days when the two hazards occurred simultaneously, and the duration of consecutive compound events (CSEs) refers to the sum of the number of days during which both hazards occurred during the two growth stages. It is the total during each kind of compound events. Our spatial maps (Figure 2 and 3) show the annual summation (1981-2014) of the frequency and duration of each compound event and didn't show the average of duration over each growing stages per year.

Given that both Review #1 (RC1.4) and Review #3 (RC3.3 b) questioned the soundness of the indicator duration, we subsequently used severity instead of duration, according to the definition of Haqiqi (Haqiqi et al., 2021), which would take into account both the duration and intensity of the disasters.

L236: Please clarify where Hunan is located by geographical lat-lon.

**RE:** The latitude and longitude range of Hunan Province is 24.38-30.08 N, 108.47-114.15 E. The latitude and longitude coordinates of these two stations in Hunan where C2R2 occurs relatively frequently are (28.6 N,112.4 E) and (29.4 N, 112.4 E).

L251-257: I couldn't understand this part. Please consider heavy modification of the text and clarification.

**RE:** This part focuses on whether when a consecutive compound events (CSEs) event occurs, the temperature and moisture behind that event may also be closely coupled, that is, whether the frequency and duration of CSEs are closely related to temperature-moisture coupling. Our results show that no clear pattern was observed between the occurrence/duration of CSEs and temperature-moisture coupling. So we conclude that there is no potential climate driver behind CSE events.

Fig 4: what is the density in the plots? And what do we learn from it?

RE: Here, the density is the density distribution of temperature-moisture Pearson-rank correlation coefficients. We tried to understand whether there is any connection between the duration of compound events (or its severity) to the growth-stage temperature-moisture coupling (as denoted by their Pearson rank Correlation Coefficients). For instance, for Figure 4(a), the density curve was derived from the correlation coefficients between three growth-stage mean temperature (T) and soil moisture (SM) for single-rice $r_{T,SM}$, over 34 years of 28 single-rice station, and in total there are 28*3=84 samples.

L263: sensitivity of PER to what for late rice? – the sentence is generally unclear.

**RE:** We think you might have asked PRE(capitation). We took the path coefficient as the relative sensitivity of $DUR$ to $T$ and $PRE$ for late-rice. The absolute values of the path coefficient indicate the extent to which the two elements, temperature and moisture, have an influence in a certain kind of compound event. In addition, we can also see how the effects of temperature and moisture differ in different types of composite events in Figure 5.

L355: consider removing the first line and directly go to the limitations you think the study has.

**RE:** Thank you! We have revised as suggested.

**References:**

Haqiqi, I., Grogan, D. S., Hertel, T. W., & Schlenker, W. (2021). Quantifying the impacts of compound extremes on agriculture. Hydrology and Earth System Sciences, 25(2), Article 2. https://doi.org/10.5194/hess-25-551-2021

Wu, H., Zhang, J., Zhang, Z., Han, J., Cao, J., Zhang, L., Luo, Y., Mei, Q., Xu, J., & Tao, F. (2023). AsiaRiceYield4km: Seasonal rice yield in Asia from 1995 to 2015. Earth System Science Data, 15(2), 791–808. https://doi.org/10.5194/essd-15-791-2023

Ye, T., Nie, J., Wang, J., Shi, P., & Wang, Z. (2015). Performance of detrending models of crop yield risk assessment: Evaluation on real and hypothetical yield data. Stochastic Environmental Research and Risk Assessment, 29(1), 109–117. https://doi.org/10.1007/s00477-014-0871-x

---

## Author Comment (AC3)

**Title: Spatiotemporal variation of growth-stage specific compound climate extremes for rice in South China: Evidence from concurrent and consecutive compound events**

**Response to Reviewer Comments (RC1):**
**'Comment on esd-2024-8', Anonymous Referee #1, 30 May 2024**

This paper investigates the combined climate extremes relevant to rice production in China. The authors analyze concurrent and consecutive compound events relevant for single- and late-rice during 1980–2014, using specific known thresholds. Examining both concurrent and consecutive extremes provides a more comprehensive picture of potential stress on rice crops. However, the manuscript would benefit from addressing some fundamental points and key concerns:

**RE:** Thank you so much for your comments and suggestions on our manuscript. We have responded to the comments and suggestions point-by-point below (in blue).

Major concerns:

**RC1.1** Sample Size Concerns: First concern is regarding the sample size of stations, highlighting the potential lack of representativeness for the entire region. Given the substantial spatial heterogeneity of soil moisture, the limited number of stations may not fully capture the diverse conditions across China.

**RE:** Thank you for the question. Indeed, the current version of our study have limited sample due to limited number of agrometeorological stations. In order to enlarge the sample size, per the suggestion from reviewer #3, we have decided to use gridded climate data of the rice cultivation areas in southern China (in this study we only focused on southern China without considering rice cultivation regions in Northeast China). We use the gridded observation climate data from CN05.1 (0.25°×0.25°) (Wu J. & Gao, 2013). This dataset has been regarded as the best choice of gridded climate forcing data in mainland China area and has been most widely used in previous studies (Li et al., 2022; Yang et al., 2017; Zhu & Yang, 2020). We use the distribution maps of single-rice (Shen et al., 2023) and late-rice (Pan et al., 2021) for year 2020 as the southern China rice growing area mask. The spatial resolution of this rice distribution data was 10 m. As one single climate grid convers many 10-m rice pixels, we selected climate grids with rice pixels ≥5% area of each 0.25°×0.25° grid. With the update, our sample sizes increased from 28 stations to 2262 grids for single-rice and from 37 stations to 1383 grids for late-rice (Fig. R1). The updated sample size would be sufficient for subsequent statistical analyses.

[Figure]

Figure R1. Comparison map of station samples and updated raster samples in the study area.

**RC1.2** Missing Yield Impact Assessment: While the paper mentions rice yield as motivation, it lacks a direct evaluation of how these compound events affect production quantities. It is necessary to incorporate an analysis of yield data to directly assess the impact of compound events on rice production. The paper's association with rice is primarily through growing season definitions, yet there is a noticeable absence of yield estimation. The justification for focusing on rice should be more explicit, particularly considering the absence of yield data.

**RE:** Thank you for your suggestions. We have tried to assessed the actual impact of climate indicators on yield (Fig. R2). Tentatively, we have firstly finished the evaluation of late rice against compound chilling-rainy events.

Here, we used AsiaRiceYield4km data (H. Wu et al., 2023) as the yield raster data, covering the period of 1995 to 2015. It is so far the dataset that provides the longest time-series covering whole China rice cultivation areas. Rice yield data with even longer time-series could only rely on the agrometeorological stations, which would again suffer from the sample size issue. To measure the impact, we followed Ye (Ye et al., 2015) by using detrended yield anomaly to remove the spatial difference in yield.

For the intensity of events, we used severity indicators based on suggestion-RC1.4. For chilling, we used the cold-degree-days of the growth stage as the severity. The cumulative deficit of average daily temperature (Tmean) ≤20°C for three or more consecutive days:

$$CDD_{stage} = \sum_{i=1}^{n} |TEM_{base} - TEM_i|$$

$CDD_{stage}$ represents the cold-degree-days for each growth stage. $i$ is the index of the day within the consecutive days that meet the condition. $TEM_i$ is the mean daily temperature value on day $i$. $TEM_{base}$ is the mean daily temperature threshold (20°C during Heading-flowering stage (stage#2) and 17°C during Grain filling stage (stage#3), according to our threshold indicated in the manuscript. $n$ is the number of consecutive days that satisfy the condition (at least 3 days).

For the impact of rainy event, we used the cumulative precipitation greater than or equal to 25 mm for three or more consecutive days. A daily 25mm rainfall was classified as the rainy in <QX/T, 468-2018, Code of Agricultural Meteorological Observations-Rice > for precipitation:

$$PDD_{stage} = \sum_{i=1}^{n} |PRE_i - PRE_{base}|$$

$PDD_{stage}$ represents the precipitation-degree-days for each growth stage. $i$ is the index of the day within the consecutive days that meet the condition. $PRE_i$ is the daily precipitation value on day $i$. $PRE_{base}$ is the daily precipitation threshold (25 mm). $n$ is the number of consecutive days that satisfy the condition (at least 3 days).

[Figure]

Figure R2. Late-rice yield responses to severity of chilling (temperature) and rainy (precipitation) variation. Color bands indicate the value of the yield anomaly.

Several interesting things could be observed from the figures:

1) There is a clear compound impact of chilling-rainy events on late rice. As severity of chilling

or rainy events increased (from the bottom left to the top right of the graphs), yield decreased. The scatters indicate a weakly convex set of isolines, indicating a weakly stronger yield impact than the linear average of single events, that said, the compound impact of having chilling-rainy together would be stronger than the linear combination of the impacts from each stressor.

2) The impact was more severe the Heading-flowering stage (stage#2) than in the Grain filling stage (stage#3), although there were much less compound events in stage #2 than in stage #3 stage. Negative yield anomaly occurred at smaller values of severity in Fig. R2(a) than that in Fig. R2(b).

Due to the limited time of writing up this response, we have not yet finished the rest part of the analyses, i.e. the impact of concurrent heat-drought events on single rice, and the consecutive events.

**RC1.3** Growing Season Definition Clarity: Specifying whether the growing season definition has fixed planting and harvest dates or adapts based on actual planting times is crucial. Sensitivity analysis to choice of dates is necessary to understand how changes in the selection of growing season start and end could influence the results.

**RE:** In the present version, we used actual planting times (differing year-by-year) as we have sufficient agrometeorological station records to do so. As we are proposing to capture the climate extremes for different phenological stages, timing really matters. For instance, the flowering stage has only 7-10 days, and the interannual variation of phenological dates could sufficiently affect the detection of climate extremes in this stage should a fixed dates be used.

**RC1.4** Intensity Metric Considerations: The current focus on number of extreme days based on thresholds might overlook the intensity of extreme events. Analyzing the magnitude of temperature or drought deviations could provide deeper insights. The metrics employed in the study center on frequency and the number of days above a threshold but fail to consider the intensity of compound events. It is important to consider the intensity, as a single day with a temperature 10°C above the threshold could have more substantial implications for agriculture than ten days with only 0.5°C above the threshold.

**RE:** We strongly agree with reviewer #1's suggestion to include an intensity indicator. The existing literature has proposed three indicators to descript an extreme event: intensity, duration, and severity (Haqiqi et al., 2021)(Fig. R3). According to this reference, we have decided to use severity in the revision, which combines both the duration and intensity of the extreme event based on the cumulative deviation from the threshold for each type of extreme event. For instance, in our impact analysis shown earlier, we have tried to use cold degree days for chilling stress, and cumulative rainfall above 25mm (daily) for rainy stress. Tentatively, they seemed capable to capture the compound impact on yield. Later, we will test on compound heat and drought. When we derive the severity of the compound event, we plan to derive the bivariate probability as the measure of intensity of the compound event, by using copulas and survival Kendall return periods approach or similar approaches.

[Figure]

dynamics of soil moisture conditions

intensity [i]

critical high

normal

time

duration [iii]

critical low

severity [iv]

intensity [ii]

A: Extreme surplus       D: Around normal*

B: Surplus       E: Deficit

C: Around normal*       F: Extreme deficit

**Figure 1.** Soil moisture dynamics within a typical growing season. Some soil moisture conditions can be harmful to crops, including excess wetness (i), moisture stress intensity (ii), duration of moisture stress (iii), and severity of soil moisture stress (iv). The around normal (*) levels can be determined by statistically examining the impacts of various intervals of soil moisture deviation from normal (seasonal mean volumetric soil moisture).

Figure R3. Reference chart for the definition of severity (https://doi.org/10.5194/hess-25-551-2021).

**RC1.5** Practical Implications and Value Added: Explicitly discussing the practical applications of the research and its contribution to existing knowledge would enhance the paper's value for the scientific community.

**RE:** Our study has three particular values.1) While most previous studies have used relative thresholds to define climate extremes, our study considers the physiological responses and thresholds of specific crops. Therefore, the results provide the most accurate view of the extreme events. 2) Crop sensitivity to climate extremes varies by growth stage and event type. Unlike previous studies conducted over the full growing season, our study carefully distinguishes between the three growth stages of rice (the jointing-booting stage (#1), the heading-flowering stage (#2), and the grain-filling stage (#3)). This study at the scale of the remaining rice stages allows us to see the differences in crop impacts of extreme events occurring at different growth stages. 3) At the sub-growth stage scale, we distinguish between multiple types of compound events (concurrent and consecutive climate extremes) that can occur. Combined with the newly added yield impact assessment section (RE1.2), our results are able to see the impact of different types of compound events on rice yields.

**References:**

Haqiqi, I., Grogan, D. S., Hertel, T. W., & Schlenker, W. (2021). Quantifying the impacts of compound extremes on agriculture. *Hydrology and Earth System Sciences*, *25*(2), Article 2. https://doi.org/10.5194/hess-25-551-2021

Li, Z., Liu, W., Ye, T., Chen, S., & Shan, H. (2022). Observed and CMIP6 simulated occurrence and intensity of compound agroclimatic extremes over maize harvested areas in China. *Weather and Climate Extremes*, *38*, 100503. https://doi.org/10.1016/j.wace.2022.100503

Pan, B., Zheng, Y., Shen, R., Ye, T., Zhao, W., Dong, J., Ma, H., & Yuan, W. (2021). High Resolution Distribution Dataset of Double-Season Paddy Rice in China. *Remote Sensing*, *13*(22), Article 22. https://doi.org/10.3390/rs13224609

Shen, R., Pan, B., Peng, Q., Dong, J., Chen, X., Zhang, X., Ye, T., Huang, J., & Yuan, W. (2023). High-resolution distribution maps of single-season rice in China from 2017 to 2022. *Earth System Science Data*, *15*(7), 3203–3222. https://doi.org/10.5194/essd-15-3203-2023

Wu, H., Zhang, J., Zhang, Z., Han, J., Cao, J., Zhang, L., Luo, Y., Mei, Q., Xu, J., & Tao, F. (2023). AsiaRiceYield4km: Seasonal rice yield in Asia from 1995 to 2015. *Earth System Science Data*, *15*(2), 791–808. https://doi.org/10.5194/essd-15-791-2023

Yang, F., Lu, H., Yang, K., He, J., Wang, W., Wright, J. S., Li, C., Han, M., & Li, Y. (2017). Evaluation of multiple forcing data sets for precipitation and shortwave radiation over major land areas of China. *Hydrology and Earth System Sciences*, *21*(11), 5805–5821. https://doi.org/10.5194/hess-21-5805-2017

Ye, T., Nie, J., Wang, J., Shi, P., & Wang, Z. (2015). Performance of detrending models of crop yield risk assessment: Evaluation on real and hypothetical yield data. *Stochastic Environmental Research and Risk Assessment*, *29*(1), 109–117. https://doi.org/10.1007/s00477-014-0871-x

Zhu, Y.-Y., & Yang, S. (2020). Evaluation of CMIP6 for historical temperature and precipitation over the Tibetan Plateau and its comparison with CMIP5. *Advances in Climate Change Research*, *11*(3), 239–251. https://doi.org/10.1016/j.accre.2020.08.001